# STREAMING ALGORITHMS FOR $\ell_p$ FLOWS AND $\ell_p$ REGRESSION

**Amit Chakrabarti**
Department of Computer Science
Dartmouth College
Hanover, NH 03755, USA
`amit.chakrabarti@dartmouth.edu`

**Jeffrey Jiang**
Department of Computer Science
Dartmouth College
Hanover, NH 03755, USA
`jeffrey.jiang@dartmouth.edu`

**David P. Woodruff**
Department of Computer Science
Carnegie Mellon University
Pittsburgh, PA 15213, USA
`dwoodruf@cs.cmu.edu`

**Taisuke Yasuda**
Department of Computer Science
Carnegie Mellon University
Pittsburgh, PA 15213, USA
`taisukey@cs.cmu.edu`

## ABSTRACT

We initiate the study of one-pass streaming algorithms for underdetermined $\ell_p$ linear regression problems of the form

$$\min_{\boldsymbol{Ax}=\boldsymbol{b}} \|\boldsymbol{x}\|_p, \qquad \text{where } \boldsymbol{A} \in \mathbb{R}^{n \times d} \text{ with } n \ll d,$$

which generalizes basis pursuit ($p = 1$) and least squares solutions to underdetermined linear systems ($p = 2$). We study the column-arrival streaming model, in which the columns of $\boldsymbol{A}$ and then the vector $\boldsymbol{b}$ are presented one by one in a stream. When $\boldsymbol{A}$ is the incidence matrix of a graph, this corresponds to an edge insertion graph stream, and the regression problem captures $\ell_p$ flows which includes transshipment ($p = 1$), electrical flows ($p = 2$), and max flow ($p = \infty$) on undirected graphs as special cases. Our goal is to design algorithms which use space much less than the entire stream, which has a length of $d$. For the task of estimating the cost of the $\ell_p$ regression problem for $p \in [2, \infty]$, we show a streaming algorithm which constructs a sparse instance supported on $\tilde{O}(\varepsilon^{-2}n)$ columns of $\boldsymbol{A}$ which approximates the cost up to a $(1 \pm \varepsilon)$ factor, which corresponds to $\tilde{O}(\varepsilon^{-2}n^2)$ bits of space in general and an $\tilde{O}(\varepsilon^{-2}n)$ space semi-streaming algorithm for constructing $\ell_p$ flow sparsifiers on graphs. This extends to $p \in (1, 2)$ with $\tilde{O}(\varepsilon^2 n^{q/2})$ columns, where $q$ is the Hölder conjugate exponent of $p$. For $p = 2$, we show that $\Omega(n^2)$ bits of space are required in general even for outputting a constant factor solution. For $p = 1$, we show that the cost cannot be estimated even to an $o(\sqrt{n})$ factor in $\text{poly}(n)$ space. On the other hand, if we are interested in outputting a solution $\boldsymbol{x}$, then we show that $(1 + \varepsilon)$-approximations require $\Omega(d)$ space for $p > 1$, and in general, $\beta$-approximations require $\tilde{\Omega}(d/\beta^{2q})$ space for $p > 1$. We complement these lower bounds with the first sublinear space upper bounds for this problem, showing that we can output a $\beta$-approximation using space only $\text{poly}(n) \cdot \tilde{O}(d/\beta^q)$ for $p > 1$, as well as a $\sqrt{n}$-approximation using $\text{poly}(n, \log d)$ space for $p = 1$.

## 1 INTRODUCTION

When faced with an *underdetermined* linear system $\boldsymbol{Ax} = \boldsymbol{b}$ for an $n \times d$ matrix $\boldsymbol{A}$ and a $n$-dimensional vector $\boldsymbol{b}$, a common approach towards obtaining useful solutions is to seek a vector $\boldsymbol{x} = [x_1, \ldots, x_d]^\top$ that minimizes some measure of cost. A popular choice is to minimize the least squares cost of $\boldsymbol{x}$, i.e., the $\ell_2$ norm of $\boldsymbol{x}$, in which case the exact minimizer can be written in closed form as

$\arg\min_{Ax=b}\|x\|_2 = A^\top(AA^\top)^{-1}b$. Another well-studied choice is to minimize the $\ell_1$ norm of $x$, also known as *basis pursuit* (Chen et al., 2001), which gives rise to sparse solutions $x$ and has been popularized in the literature of compressed sensing and sparse recovery.

The minimum $\ell_1$ and $\ell_2$ norm solutions when $A, b$ are training data and labels, respectively, have been of interest to understanding the double descent phenomenon: when machine learning models exhibit multiple phases of decreasing risk (or expected loss) as model complexity increases (Hastie et al., 2022; Bartlett et al., 2020; Li & Wei, 2021). Less theory is developed about regression models in this underconstrained regime as it diverges from the traditional bias-variance U-shape risk curve. Following this framework, these interpolating models should contain very high risk as it drastically "overfits" to train data. Nonetheless, the aforementioned works demonstrate both empirical and theoretical results about the extraordinary ability of these minimum norm interpolators to generalize well on unseen data. This motivates their study in models for processing large data sets.

When $A$ is the incidence matrix of a graph with $n$ vertices and $d$ edges, i.e., $A_{u,e} = -A_{v,e} = 1$ whenever $e$ is an edge from a vertex $u$ to $v$, then solutions $x$ to the linear system $Ax = b$ correspond to *flows* that respect a given set of *flow conservation constraints* or *demands* specified by the vector $b$. In this case, the problem

$$\min_{Ax=b}\|x\|_p \qquad (1)$$

is known as the *p-norm flow problem* and has applications to graph clustering (Liu & Gleich, 2020; Fountoulakis et al., 2020), network science (Kalantari et al., 2008), and captures transshipment ($p = 1$), electrical flows ($p = 2$), and max flow ($p = \infty$) as special cases. In general, problem (1) is known as the $\ell_p$ regression problem, and algorithms for solving it, both for graphs and for general matrices, have recently been a topic of intense study (Bubeck et al., 2018; Ene & Vladu, 2019; Adil et al., 2019b;a; Adil & Sachdeva, 2020; Adil et al., 2021; Jambulapati et al., 2022; 2024).

It is useful to generalize problem (1) by associating a weight $c_j$ with each column $j \in [d]$. This captures $p$-norm flows with associated *capacities*. In this setting, we wish to solve the optimization problem given by

$$\min_{Ax=b}\|C^{-1}x\|_p \qquad (2)$$

for $C = \mathrm{diag}(c_1, \ldots, c_d)$. If $c_j = 0$ for some $j \in [d]$, then we say the objective $\|C^{-1}x\|_p$ is infinite, unless $x_j = 0$.

In many underdetermined linear systems, the number $d$ of columns of $A$ far exceeds the number $n$ of rows. In this regime, it may be unreasonable to store the entire matrix $A$ in memory to solve the $\ell_p$ regression problem. This motivates a streaming model of computation for this problem, in which the algorithm only accesses a small portion of the input at a time in a stream of updates, and the goal is to solve the problem using memory that is much smaller than the length $d$ of the stream. In this work, we study the *column-arrival streaming model*, defined below.

**Definition 1.1** (Column-arrival streaming model)**.** Consider an instance of problem (2) given by $C = \mathrm{diag}(c_1, \ldots, c_d) \in \mathbb{R}^{d \times d}$, $A \in \mathbb{R}^{n \times d}$, and $b \in \mathbb{R}^n$. We say that this instance is presented in a *column-arrival stream* if we receive $d + 1$ updates in a stream in an arbitrary order, where each update consists of either a pair $(a^j, c_j)$ consisting of a column $a^j$ of $A$ and an associated weight $c_j$, or the vector $b$. In this work, we focus on algorithms which make only *one pass* through the data stream.

Special cases of the general streaming underdetermined $\ell_p$ regression problems have been studied by a number of works in the literature of streaming algorithms. For $p = 2$, the work of Bartan & Pilanci (2023) gives an algorithm for estimating the cost of the regression problem by using sketching techniques. When $A$ corresponds to the incidence matrix of a graph, this model corresponds to an *edge insertion graph stream* (McGregor, 2014), and the corresponding graph streaming problems for transshipment (Becker et al., 2021), electrical flows (Kelner & Levin, 2013; Kapralov et al., 2014; Cohen et al., 2016), and max flow (Assadi et al., 2019) have received much attention in the literature. The literature on graph streaming problems related to the max flow problem, such as reachability (Guruswami & Onak, 2013; Assadi & Raz, 2020; Chen et al., 2021) and maximum bipartite matching (Feigenbaum et al., 2005; McGregor, 2005; Ahn & Guha, 2013; Crouch & Stubbs, 2014; Paz & Schwartzman, 2017; Assadi et al., 2017; Kapralov, 2021) is even more vast.

In the study of streaming algorithms, we are typically interested in minimizing the space used by the algorithm. In our algorithms, we will allow for $A$ to take real values and bound the space complexity

in terms of the number of columns of $A$ stored and additional words of space, and also give bit complexity bounds when the entries of $A$ are bounded integers. In our lower bounds, we will prove bit complexity lower bounds in the latter complexity model.

## 1.1 OUR CONTRIBUTIONS

We design streaming algorithms for problem (2), handling a wide range of parameter settings. We also prove several corresponding lower bounds. We consider both the *cost estimation* problem, where the desired output is a real number that approximates the minimum $\ell_p$ norm, as well as the harder *vector-valued* problem, where the desired output is an approximate minimizer vector $x$. We work in the setting $n \ll d$. In stating our results, we use the notations $\tilde{O}(\cdot)$ and $\tilde{\Omega}(\cdot)$ to suppress factors polylogarithmic in $n$ and $d$.

### 1.1.1 ESTIMATING THE MINIMUM COST

For the most basic setting of problem (1), where $p = 2$, there is a well-known closed form expression for the minimizer vector: $x^* = A^\top M^{-1} b$, where $M := AA^\top$. Therefore, the minimum cost can be computed in $O(n^2)$ words of space, assuming real arithmetic. The method is straightforward: maintain the matrix $M$ exactly, using the fact that each stream update—i.e., each new column of $A$—makes a rank-1 additive update to $M$. At the end of the stream, return the solution $\|A^\top M^{-1} b\|_2 = (b^\top M^{-1} b)^{1/2}$.

Notice that the space bound $O(n^2)$ is sublinear for $d = \omega(n)$. We shall eventually prove that this quadratic dependence on $n$ is *optimal*, even for returning a constant-factor approximation to the minimum cost. This essentially settles the streaming complexity of $\ell_p$ regression for $p = 2$. However, it is unclear how these results generalize to the setting of $p \neq 2$, when no closed form solutions are available for problem (1).

Our algorithms for more general $p \in (1, \infty]$ provide approximations to the cost of $\ell_p$ regression, up to a $(1 + \varepsilon)$ factor. They are based on the streaming construction of a notion of $\ell_p$ *flow sparsifiers*, which we define as follows.

**Definition 1.2** (Flow sparsifier). Let $C \in \mathbb{R}^{d \times d}$ be a diagonal matrix, $A \in \mathbb{R}^{n \times d}$, and $b \in \mathbb{R}^n$. Let $p \in [1, \infty]$. Then, a diagonal matrix $S$ is a $\beta$-*approximate* $\ell_p$ *flow sparsifier* with sparsity $\mathsf{nnz}(S)$ if

$$\min_{Ax=b} \|C^{-1} x\|_p \leq \min_{Ax=b} \|S^{-1} C^{-1} x\|_p \leq \beta \cdot \min_{Ax=b} \|C^{-1} x\|_p.$$

In the graph setting, when $A$ is the incidence matrix of a graph, our notion of a flow sparsifier corresponds to a weighted subgraph whose $\ell_p$ flow cost approximates that of the original graph. This generalizes the notion of spectral sparsifiers (Spielman & Teng, 2011), which correspond to the case $p = 2$. Note that this differs from another line of work on "flow sparsifiers" that focuses on *vertex sparsification* for preserving the cost of flows when there are only a small number of terminals of interest (Leighton & Moitra, 2010; Andoni et al., 2014; Krauthgamer & Mosenzon, 2023; Chen & Tan, 2024), whereas we construct *edge* sparsifiers with general vectors $b$, similar to a notion of flow sparsifiers used in Sherman (2013); Kelner et al. (2014).

Due to the work of Cohen et al. (2016), it is known how to construct electrical flow sparsifiers even in an *online fashion*, that is, the sparsifier can be constructed in one pass over an insertion stream of edges with the guarantee the edges are only ever kept and never thrown away.

To state the space requirements of our algorithm in full generality, we need the following notion of an *online condition number* (Cohen et al., 2016; Woodruff & Yasuda, 2022).

**Definition 1.3** (Online condition number). Let $A \in \mathbb{R}^{n \times d}$. Then, the *online condition number* $\kappa^{\mathsf{OL}}(A)$ is defined as $\kappa^{\mathsf{OL}}(A) := \|A\| \max_{j \in [d]} \|A_j^-\|$, where $A_j$ denotes the $n \times j$ submatrix of $A$ formed by its first $j$ columns, $A_j^-$ denotes the Moore–Penrose pseudoinverse of this matrix, and $\|\cdot\|$ denotes the matrix 2-norm.

This brings us to our main algorithmic result for constructing $\ell_p$ flow sparsifiers, in particular solving the cost estimation version of problem (2).

**Theorem 1.** *Let $p \in (1, \infty]$ and let $q = p/(p-1) \in [1, \infty)$ be its Hölder conjugate exponent. There is an algorithm that reads $A \in \mathbb{R}^{n \times d}$ and the diagonal matrix $C \in \mathbb{R}^{d \times d}$ in a column-arrival stream*

*(Definition 1.1) and, with probability at least $1 - \delta$, outputs a $(1 + \varepsilon)$-approximate $\ell_p$ flow sparsifier (Definition 1.2). Furthermore, the algorithm stores at most s columns of $\mathbf{AC}$, with*

$$s = O\left(\varepsilon^{-2} n^{\max\{1, q/2\}}\right) \operatorname{poly} \log(d, \kappa^{\mathsf{OL}}(\mathbf{AC}), 1/\delta)$$

*and at most $O(n^2)$ additional words of space if $\mathbf{A}$ has real entries, and*

$$s = O\left(\varepsilon^{-2} n^{\max\{1, q/2\}}\right) \operatorname{poly} \log(d, 1/\delta)$$

*and at most $O(n^2 \log d)$ bits of additional space if $\mathbf{AC}$ has integer entries bounded by $\pm \operatorname{poly}(d)$. Furthermore, the s columns of $\mathbf{AC}$ are stored in an* online *fashion, that is, the columns are selected irrevocably and are not thrown away throughout the stream.*

A polylogarithmic dependence on the online condition number is typical of results in the literature of online numerical linear algebra, and is known to be necessary in general (Cohen et al., 2016). For $p \in [2, \infty]$, our algorithms store only $\tilde{O}(\varepsilon^{-2} n)$ columns of $\mathbf{A}$, which corresponds to an $\tilde{O}(\varepsilon^{-2} n)$ space algorithm—i.e., a "semi-streaming" algorithm—in the setting of $\ell_p$ flows on undirected graphs. Theorem 1 generalizes results on spectral sparsification in the semi-streaming setting (Kelner & Levin, 2013; Kapralov et al., 2014; Cohen et al., 2016), which corresponds to the case of $p = 2$. We also note that Theorem 1 separates the space complexity of max flow on undirected graphs from that on directed graphs. Indeed, on directed graphs, even the reachability problem, which is a special case of a directed max flow, requires an almost quadratic $\Omega(n^{2-o(1)})$ bits of space (Guruswami & Onak, 2013; Assadi & Raz, 2020; Chen et al., 2021), even with multiple passes.

We now turn to lower bound results. These results all have the following structure. Suppose that there exists a column-arrival streaming algorithm for problem (1) that uses at most $B$ bits of space and, with probability at least $2/3$, outputs a "good estimate" $c$ to the cost $\min_{\mathbf{Ax}=\mathbf{b}} \|\mathbf{x}\|_p$. Then, in particular, there must exist randomized algorithms $\mathscr{A}$ and $\mathscr{B}$ such that, for any $\mathbf{A} \in \mathbb{R}^{n \times d}$, $\mathscr{A}$ produces an at-most-$B$-bit message $M = \mathscr{A}(\mathbf{A})$ so that, for any $\mathbf{b} \in \mathbb{R}^n$, $\mathscr{B}$ outputs the good estimate $c = \mathscr{B}(M, \mathbf{b})$ as a function of $M$ and $\mathbf{b}$. The proofs of our lower bounds apply to this more relaxed setting, effectively allowing an unlimited amount of space to process the columns of $\mathbf{A}$ as they are streamed in and only enforcing a space limitation before $\mathbf{b}$ is revealed.

First, we consider the algorithmically "easy" case of problem (1), when $p = 2$. We remarked above that the folklore $O(n^2)$-space algorithm is optimal in its dependence on $n$. Formally, we establish the following result.

**Theorem 2.** *Fix $p = 2$. There is an absolute constant $\alpha > 0$ such that any column-arrival streaming algorithm that, with probability at least $2/3$, computes a $(1 + \alpha)$-approximation to the cost of problem (1) requires $\Omega(n^2)$ bits of space.*

Next, we consider $p = 1$ in problem (1). In this case, the Hölder conjugate $q = \infty$ and thus Theorem 1 does not give a $(1 \pm \varepsilon)$-approximation algorithm. We show that this is inherent by proving the following lower bound.

**Theorem 3.** *Fix $p = 1$ and let $D \geq 1$ be arbitrary. There is a constant $C_D > 0$ such that any column-arrival streaming algorithm that, with probability at least $2/3$, computes an estimate $c$ with $c \leq \min_{\mathbf{Ax}=\mathbf{b}} \|\mathbf{x}\|_1 < (\sqrt{n}/C_D)c$ requires $\Omega(n^D)$ bits of space. This result applies even when all entries of the input matrix $\mathbf{A}$ lie in $\{\pm 1\}$.*

In other words, for $p = 1$, there is no $\operatorname{poly}(n)$ space algorithm for estimating the cost of $\ell_1$ regression, even up to a factor of $o(\sqrt{n})$. In contrast, we shall soon present a nearly matching algorithm that uses just $O(n) \operatorname{poly} \log d$ bits of space to output an actual solution *vector* $\mathbf{x} \in \mathbb{R}^d$ that achieves a $\sqrt{n}$-factor approximation.

Finally, one can take $p = 0$ in problem (1), with the natural interpretation that $\|\mathbf{x}\|_0 = \mathsf{nnz}(\mathbf{x})$. This setting does not admit a sublinear space solution, even with $n$ as small as 2, which we note below.

**Theorem 4.** *Fix $p = 0$ and take $n = 2$. Any column-arrival streaming algorithm that, with probability at least $2/3$, outputs a 2-approximation to the cost minimum cost in problem (1) requires $\Omega(d)$ bits of space.*

Table 1 summarizes the above results—both upper and lower bounds—for estimating the cost of $\ell_p$ regression.

| Range of $p$ | Distortion | Space complexity (bits of space) | Reference |
|:---:|:---:|:---:|:---:|
| $p = 2$ | $1$ | $\tilde{O}(n^2)$ | Folklore |
| $p \in (2, \infty]$ | $(1+\varepsilon)$ | $\tilde{O}(\varepsilon^{-2} n^2)$ | Theorem 1 |
| $p \in (1, 2)$ | $(1+\varepsilon)$ | $\tilde{O}(\varepsilon^{-2} n^{q/2+1})$ | Theorem 1 |
| $p = 2$ | $(1+\varepsilon)$ | $\Omega(n^2)$ | Theorem 2 |
| $p = 1$ | $o(\sqrt{n})$ | $n^{\omega(1)}$ | Theorem 3 |
| $p = 0$ | $2$ | $\Omega(d)$ | Theorem 4 |

Table 1: Space complexity of estimating the cost of $\ell_p$ regression, with $q := p/(p-1)$

### 1.1.2 OUTPUTTING A GOOD SOLUTION

The streaming algorithm for $\ell_p$ flows mentioned in Section 1.1.1 only output a *scalar*, approximating the cost of the regression problem. More generally, one would want an actual *solution vector* $\boldsymbol{x} \in \mathbb{R}^d$ with small $\ell_p$ norm that satisfies $\boldsymbol{A}\boldsymbol{x} = \boldsymbol{b}$. At first glance, this may feel like asking for too much: after all, $\boldsymbol{x}$ is $d$-dimensional, which would seem to necessitate at least $\Omega(d)$ space, thus precluding any sublinear space streaming algorithm. However, for $p = 1$, one of the primary reasons for solving the basis pursuit problem $\min_{\boldsymbol{A}\boldsymbol{x}=\boldsymbol{b}} \|\boldsymbol{x}\|_1$ is to identify sparse solutions $\boldsymbol{x}$, which require much less than $d$ space to specify. Furthermore, it may be possible to specify solutions in less than $d$ space if we allow for some large distortion (i.e., approximation factor) in the solution, say $d^{0.1}$.

Thus, it *is* worthwhile to look for algorithms that can output a solution vector $\boldsymbol{x}$, especially if we allow for approximation errors as large as $\text{poly}(n, d)$. We do design such an algorithm. Before discussing it, we bring up a couple of related lower bounds that we prove; these serve to set the context for the algorithm. Our lower bounds hold even in the very special setting of $n = 1$, so the matrix $\boldsymbol{A}$ becomes a row vector $\boldsymbol{a}$ and the vector $\boldsymbol{b}$ becomes a scalar $b$. Furthermore, the entries of $\boldsymbol{a}$ can be restricted to $\{\pm 1\}$ and the scalar $b$ on the right-hand side of problem (1) can be fixed to $d$.

As with the lower bounds in Section 1.1.1, the next two lower bounds hold in the more relaxed setting where $\boldsymbol{a}$ can be processed by a randomized algorithm $\mathscr{A}$ using unlimited space, resulting in a $B$-bit message $M = \mathscr{A}(\boldsymbol{a})$. Another randomized algorithm $\mathscr{B}$ must then produce a good output vector $\hat{\boldsymbol{x}} = \mathscr{B}(M)$, based on $M$, succeeding with high probability. Clearly, this setting is a relaxation of a column-arrival streaming algorithm that uses $B$ bits of space.

**Theorem 5.** *Let $p \in (1, \infty]$ and let $q = p/(p-1) \in [1, \infty)$ be its Hölder conjugate exponent. Let $\varepsilon \in (0, 1/(8q))$ and $d \in \mathbb{N}$. Any randomized algorithm that computes a $B$-bit summary of $\boldsymbol{a} \in \{\pm 1\}^d$ from which $\hat{\boldsymbol{x}} \in \mathbb{R}^d$ can be produced such that, with probability at least $2/3$, we have $\langle \boldsymbol{a}, \hat{\boldsymbol{x}} \rangle = d$ and $\|\hat{\boldsymbol{x}}\|_p \le (1+\varepsilon) \min_{\langle \boldsymbol{a}, \boldsymbol{x} \rangle = d} \|\boldsymbol{x}\|_p$ requires $B = \Omega(d)$.*

In other words, for $p > 1$, there is no algorithm using less than $d$ space that can output a $(1+\varepsilon)$-approximate solution for $\varepsilon \le 1/(8q)$.

We also obtain a lower bound in the setting of large distortions, showing that for a $\beta$-factor approximation, the streaming algorithm must use at least $\tilde{\Omega}(d/\beta^{2q})$ bits of space, provided that $\beta^{3q} \ll d$. In particular, for $p > 1$, it is in fact not possible to output a solution $\boldsymbol{x}$ in $\text{poly}(n, \log d)$ space unless the approximation factor is at least $\text{poly}(d)$.

**Theorem 6.** *Let $p \in (1, \infty]$ and let $q = p/(p-1) \in [1, \infty)$ be its Hölder conjugate exponent. Let $\beta$ be a distortion parameter such that $(\beta \log d)^{3q} = cd$ for a sufficiently small universal constant $c$. Then any randomized algorithm that computes a $B$-bit summary of $\boldsymbol{a} \in \{\pm 1\}^d$ from which $\hat{\boldsymbol{x}} \in \mathbb{R}^d$ can be produced such that, with probability at least $1 - 1/O(\beta \log d)^q$, we have $\langle \boldsymbol{a}, \hat{\boldsymbol{x}} \rangle = d$ and $\|\hat{\boldsymbol{x}}\|_p \le \beta \cdot \min_{\langle \boldsymbol{a}, \boldsymbol{x} \rangle = d} \|\boldsymbol{x}\|_p$ requires $B = \Omega(d/(\beta \log d)^{2q})$.*

We prove both of these lower bounds in Section 4. The high-accuracy lower bound of Theorem 5 follows from a relatively simple reduction to one-pass streaming lower bounds for the INDEX communication problem (Kremer et al., 1995). On the other hand, Theorem 6 is our most technically advanced lower bound result, requiring additional techniques in order to extract information about $\boldsymbol{a}$ from a $\beta$-approximate solution for large $\beta$. In particular, our lower bound argument involves classifying the entries of a $\beta$-approximate solution $\boldsymbol{x}$ according to their contribution towards partitioning

the coordinates into comparable classes. We then apply conditioning on an additional short string of advice to construct an estimator that extracts many bits of information about the input $a$ from the solution $x$.

Turning to upper bounds, we give a new algorithm for the general $\ell_p$ regression problem (2) that runs in a strongly sublinear $d^{1-\Omega(1)}$ amount of space and outputs a solution vector achieving $d^{1-\Omega(1)}$ distortion. Furthermore, in the case of $p = 1$, our algorithm outputs a solution with distortion at most $\sqrt{n}$ with space complexity $\text{poly}(n, \log d)$, which explains why the lower bounds of Theorems 5 and 6 do not apply when $p = 1$.

Below, we write $A|^S$ to denote the $n \times |S|$ submatrix of $A$ with columns indexed by $S \subseteq [d]$.

**Theorem 7.** *Let $p \in [1, \infty]$ and let $q = p/(p-1) \in [1, \infty]$ be its Hölder conjugate exponent. There is an algorithm that reads $A \in \mathbb{R}^{n \times d}$ and the diagonal matrix $C \in \mathbb{R}^{d \times d}$ in a column-arrival stream and stores a subset $S$ of at most $O(sd/\beta^q)$ columns of $A$ and entries of $C$ for $p > 1$, and $O(s)$ columns of $A$ and entries of $C$ for $p = 1$, such that*

$$\min_{A|^S x = b} \|C^{-1} x\|_p \leq \begin{cases} \beta \cdot \min_{Ax=b} \|C^{-1} x\|_p, & \text{when } p \geq 2, \\ n^{1/p - 1/2} \beta \cdot \min_{Ax=b} \|C^{-1} x\|_p, & \text{when } p < 2, \\ n^{1/2} \cdot \min_{Ax=b} \|C^{-1} x\|_p, & \text{when } p = 1, \end{cases} \tag{3}$$

*where $s = O(n \log(d\kappa^{\mathsf{OL}}(AC)))$ if $AC$ has real entries, and $s = O(n \log d)$ if $AC$ is an integer matrix with entries bounded in absolute value by $\text{poly}(d)$. Furthermore, the $s$ columns of $AC$ are stored in an* online *fashion, that is, the columns are selected irrevocably and are not thrown away throughout the stream.*

As the statement of the above theorem suggests, our algorithm is based on a *column subset selection* approach. It in fact outputs a *sparse solution* that approximately solves the $\ell_p$ regression problem. Furthermore, the trade-off between the approximation factor $\beta$ and the sparsity of this solution improves as $p \to 1$, which affirms the intuition that minimizing the $\ell_p$ norm for $p$ closer to 1 yields sparser solutions. Our algorithm uses the idea of *well-conditioned spanning sets*, which is a subset of the columns of $A$ such that all other columns can be written as a linear combination of this subset with small coefficients, and we show that such subsets can in fact be constructed in a streaming fashion by using a streaming Löwner–John ellipsoid algorithm of Woodruff & Yasuda (2022).

Table 2 summarizes our upper and lower bounds for the problem of outputting a solution vector.

| Range of $p$ | Distortion | Space complexity (bits of space) | Theorem |
|---|---|---|---|
| $p \in (1, \infty]$ | $(1 + \varepsilon)$ | $\Omega(d)$ | Theorem 5 |
| $p \in (1, \infty]$ | $\beta$ | $\tilde{\Omega}(d/\beta^{2q})$ | Theorem 6 |
| $p \in (2, \infty]$ | $\beta$ | $n^2 \cdot \tilde{O}(d/\beta^q)$ | Theorem 7 |
| $p \in (1, 2)$ | $n^{1/p - 1/2}\beta$ | $n^2 \cdot \tilde{O}(d/\beta^q)$ | Theorem 7 |
| $p = 1$ | $n^{1/2}$ | $n^2 \cdot \text{poly} \log d$ | Theorem 7 |

Table 2: Space complexity of outputting a solution vector for $\ell_p$ regression, with $q := p/(p-1)$

## 1.2 Open Questions

We have initiated the study of equality-constrained norm minimization in the streaming setting, and there are a number of natural questions that remain unresolved. Our first question is on closing the gap between our upper and lower bounds for streaming $\ell_p$ regression algorithms that output an approximate solution $x$ with large distortion. We conjecture that our lower bound is tight and raise the question of tightening the upper bound.

**Question 1.4.** *Is there a column-arrival streaming algorithm which outputs a solution $\hat{x}$ satisfying $A\hat{x} = b$ and $\|\hat{x}\|_p \leq \beta \cdot \min_{Ax=b} \|x\|_p$ using space at most $\text{poly}(n) \cdot \tilde{O}(d/\beta^{2q})$?*

Furthermore, in this work, we have focused on the setting of one-pass algorithms, and we leave the question of studying algorithms and lower bounds for multi-pass algorithms for future work.

**Question 1.5.** *What is the space complexity of $\ell_p$ regression in the column-arrival streaming model for algorithms that make multiple passes over the data stream?*

Lastly, our study has been mostly focused on the setting of $\ell_p$ regression, and we have left open the possibility of obtaining better algorithms and lower bounds in the more specific setting of $\ell_p$ flows on undirected graphs. For instance, we already know that our lower bound of Theorem 3 can be circumvented by using streaming algorithms for constructing spanners (Althöfer et al., 1993; Feigenbaum et al., 2008; Baswana, 2008).

**Question 1.6.** *What is the space complexity of $\ell_p$ flows on undirected graphs in the edge insertion graph streaming model? What about for multi-pass algorithms?*

## 2   FLOW SPARSIFIERS VIA ONLINE LEWIS WEIGHT SAMPLING

In this section, we establish our first algorithmic result (Theorem 1). Our starting point for constructing $\ell_p$ flow sparsifiers is the following duality lemma; we provide a proof in Section B for completeness.

**Lemma 2.1** (Strong duality for $\ell_p$ regression). *Let $p \in [1,\infty]$, $q = p/(p-1) \in [1,\infty]$, and $C$, $A$ and $b$ be an instance of problem (2) such that $Ax = b$ is feasible. Then*

$$\min_{Ax=b} \|C^{-1}x\|_p = \max_{\|CA^\top y\|_q \leq 1} y^\top b.$$

*Proof of Theorem 1.* Suppose $S \in \mathbb{R}^{d \times d}$ is a diagonal matrix with $\|SCA^\top y\|_q = (1 \pm \varepsilon)\|CA^\top y\|_q$ simultaneously for every $y \in \mathbb{R}^n$. (This notion of $\ell_q$ norm preservation for a subspace is known as an $\ell_q$ *subspace embedding*.) From the above duality lemma, it follows that a modified dual problem with capacities $SC$ approximates the original dual up to a $(1 \pm \varepsilon)$ factor. To be precise,

$$\max_{\|SCA^\top y\|_q \leq 1} y^\top b = (1 \pm \varepsilon) \max_{\|CA^\top y\|_q \leq 1} y^\top b.$$

In turn, by applying strong duality on both sides, we have that

$$\min_{Ax=b} \|S^{-1}C^{-1}x\|_p = (1 \pm \varepsilon) \min_{Ax=b} \|C^{-1}x\|_p. \tag{4}$$

That is, the $\ell_p$ flow problem with capacities $SC$ approximates the one capacities $C$.

It remains to construct a sparse reweighting matrix $S$ such that $\|SCA^\top y\|_q = (1 \pm \varepsilon)\|CA^\top y\|_q$ simultaneously for every $y \in \mathbb{R}^n$. This can be done via a random sampling technique known as $\ell_p$ *Lewis weight sampling* (Cohen & Peng, 2015; Woodruff & Yasuda, 2023a), which shows that it is possible to construct $S$ with

$$\mathsf{nnz}(S) = \tilde{O}\big(\varepsilon^{-2}n^{\max\{1,q/2\}}\big). \tag{5}$$

In fact, this construction can be done even in the streaming setting via *online $\ell_p$ Lewis weight sampling* (Woodruff & Yasuda, 2023a), up to a small overhead, which is a factor polylogarithmic in some natural parameters. To make this precise, we recall a result of Woodruff & Yasuda (2023a) on constructing $\ell_p$ subspace embeddings: they show how to obtain the guarantee $\|SBy\|_q = (1 \pm \varepsilon)\|By\|_q$ for every $y \in \mathbb{R}^n$ when $B \in \mathbb{R}^{d \times n}$ is presented in a *row-arrival* stream. Fortunately, this corresponds to a column-arrival stream for $A$ when $B = A^\top$. Their result, translated into a column-arrival setting, is as follows.

**Fact 2.2** (Online $\ell_p$ Lewis weight sampling, Theorem 3.8 of Woodruff & Yasuda (2023a)). *Let $q \in [1,\infty)$. Let $A \in \mathbb{R}^{n \times d}$. Then, there is a column-arrival streaming algorithm which, with probability at least $1 - \delta$, outputs a sampling matrix $S$ which stores at most $s$ rows such that for all $y \in \mathbb{R}^n$, $\|SA^\top y\|_q = (1 \pm \varepsilon)\|A^\top y\|_q$ with*

$$s = O(\varepsilon^{-2}n^{\max\{1,q/2\}})\operatorname{poly}\log(d, \kappa^{\mathsf{OL}}(A), 1/\delta)$$

*and at most $O(n^2)$ additional words of space if $A$ has real entries, and*

$$s = O(\varepsilon^{-2}n^{\max\{1,q/2\}})\operatorname{poly}\log(d, 1/\delta)$$

*and at most $O(n^2 \log d)$ bits of additional space if $A$ has integer entries bounded by $\pm\operatorname{poly}(d)$.*

Combining the above result with the duality-based derivation of eq. (4), we obtain our claimed algorithm for constructing $\ell_p$ flow sparsifiers. This completes the proof of Theorem 1 $\qquad\square$

**Remark 2.3.** We note that for $\ell_p$ regression for $p \in [2, \infty]$, our duality-based approach for estimating the regression cost is also compatible with the *turnstile streaming model*, in which the matrix $\boldsymbol{A}$ undergoes entrywise updates of the form $\boldsymbol{A}(i, j) \leftarrow \boldsymbol{A}(i, j) + \Delta$ for some update $\Delta$ which can be positive or negative. Indeed, a long line of work has established algorithms for recovering a sketch $\tilde{\boldsymbol{A}}$ such that $\|\tilde{\boldsymbol{A}}^\top \boldsymbol{y}\|_q = (1 \pm \varepsilon)\|\boldsymbol{A}^\top \boldsymbol{y}\|_q$ simultaneously for every $\boldsymbol{y} \in \mathbb{R}^n$ using only $\text{poly}(n, \varepsilon^{-1}, \log d)$ bits of space (Sohler & Woodruff, 2011; Meng & Mahoney, 2013; Woodruff & Zhang, 2013; Wang & Woodruff, 2019; Li et al., 2021; Woodruff & Yasuda, 2023b; Mai et al., 2023; Munteanu & Omlor, 2024). It is also possible to get strongly sublinear in $d$ space for $p \in (1, 2)$ (i.e., $q \in (2, \infty)$) (Woodruff & Zhang, 2013). Unfortunately, these techniques yield dense approximations and thus typically do not give subquadratic space algorithms in the setting of graph streams.

**Remark 2.4.** If we give up the requirement that the columns are selected in an online fashion, then we can in fact remove the $\log \kappa^{\text{OL}}(\boldsymbol{A})$ dependence from this result by using the merge-and-reduce technique to compute $\ell_p$ subspace embeddings (c.f. Braverman et al. (2020)).

## 3 OUTPUTTING A GOOD SOLUTION IN SUBLINEAR SPACE

Here, we prove our second algorithmic result (Theorem 7). Recall that $\boldsymbol{A}|^S$ denotes the $n \times |S|$ submatrix of $\boldsymbol{A}$ with columns indexed by $S \subseteq [d]$.

### 3.1 STREAMING WELL-CONDITIONED SPANNING SUBSETS

We first note that the online John ellipsoid algorithm of Woodruff & Yasuda (2022) can be used to design a streaming algorithm for identifying a small subset of rows such that every other row can be written as a linear combination of the subset with small coefficients.

**Definition 3.1** (Well-conditioned spanning subsets). Let $\boldsymbol{A} \in \mathbb{R}^{n \times d}$. Then, a subset $S \subseteq [d]$ of the columns of $\boldsymbol{A}$ is a *well-conditioned spanning subset* if for every $j \in [d]$, there exists $\boldsymbol{y} \in \mathbb{R}^{|S|}$ such that $\boldsymbol{A}|^S \boldsymbol{y} = \boldsymbol{a}^j$ and $\|\boldsymbol{y}\|_2^2 \leq 1$.

Algorithms for computing well-conditioned spanning subsets have been studied by Knuth (1985); Woodruff & Yasuda (2023b); Bhaskara et al. (2023). The following result gives an efficient construction of well-conditioned spanning subsets in a column-arrival stream.

**Lemma 3.2** (Streaming well-conditioned spanning subsets). *Let $S \subseteq [d]$ be constructed in a column-arrival stream by adding the column $\boldsymbol{a}^j$ to $S$ whenever $(\boldsymbol{a}^j)^\top (\boldsymbol{A}|^S (\boldsymbol{A}|^S)^\top)^- \boldsymbol{a}^j \geq 1$. Then, $S$ is a well-conditioned spanning subset. Furthermore, $|S| = O(n \log(d \kappa^{\text{OL}}(\boldsymbol{A})))$ where $\kappa^{\text{OL}}(\boldsymbol{A})$ is the online condition number of $\boldsymbol{A}$ (see Definition 1.3), and $|S| = O(n \log d)$ if $\boldsymbol{A}$ is an integer matrix with entries bounded by $\text{poly}(d)$.*

*Proof.* The bound on the size of $S$ is given by Woodruff & Yasuda (2022) using bounds on sum of online leverage scores (Cohen et al., 2016) so it remains to argue the correctness, which is inspired by an argument of Woodruff & Yasuda (2023b). If $j \in S$, then we can simply take $\boldsymbol{y}$ to be the standard basis vector corresponding to this index. Otherwise, we have that $(\boldsymbol{a}^j)^\top (\boldsymbol{A}|^S (\boldsymbol{A}|^S)^\top)^- \boldsymbol{a}^j < 1$, which means that $\boldsymbol{y} = (\boldsymbol{A}|^S)^- \boldsymbol{a}^j$ is a set of coefficients such that $\boldsymbol{A}|^S \boldsymbol{y} = \boldsymbol{a}^j$ with $\ell_2$ norm at most 1. $\qquad\square$

The following lemma uses well-conditioned spanning subsets to sparsify linear combinations.

**Lemma 3.3** (Sparsifying linear combinations). *Let $p \in [1, \infty]$. Let $\boldsymbol{C} \in \mathbb{R}^{d \times d}$ be a diagonal matrix, $\boldsymbol{A} \in \mathbb{R}^{n \times d}$, and let $S \subseteq [d]$ be a well-conditioned subset of the columns of $\boldsymbol{AC}$. Then, for every $\boldsymbol{x} \in \mathbb{R}^d$, there is a $\boldsymbol{z} \in \mathbb{R}^d$ such that $\text{supp}(\boldsymbol{z}) \subseteq S$, $\boldsymbol{Ax} = \boldsymbol{Az}$, and*

$$\|\boldsymbol{z}\|_p \leq \begin{cases} d^{1-1/p}\|\boldsymbol{x}\|_p, & \text{when } p > 2, \\ n^{1/p-1/2}d^{1-1/p}\|\boldsymbol{x}\|_p, & \text{when } p \leq 2. \end{cases}$$

*Proof.* We may assume that $d \geq n$, since otherwise we can take $\boldsymbol{z} = \boldsymbol{x}$. By the definition of well-conditioned spanning subsets, we can write $\boldsymbol{A} = (\boldsymbol{AC})|^S (\boldsymbol{C}|^S)^{-1} \boldsymbol{Y} = \boldsymbol{A}|^S \boldsymbol{Y}$ where $\boldsymbol{A}|^S$ denotes the

columns of $A$ indexed by $S$ and $Y$ is an $|S| \times d$ matrix of coefficients with $\|(C|^S)^{-1} Y e^{(j)}\|_2^2 \leq 1$ for every column $j \in [d]$. Then, $Ax = A|^S Y x$ so padding the vector $Yx$ with zeros gives our vector $z$ that is supported on $S$. Furthermore,

$$\|C^{-1}z\|_p = \|(C|^S)^{-1} Y x\|_p \leq \sum_{j=1}^d \|(C|^S)^{-1} Y e^{(j)}\|_p |x_j| \qquad \text{triangle inequality}$$

$$\leq n^{\max\{(1/p-1/2),0\}} \|x\|_1 \leq n^{\max\{(1/p-1/2),0\}} d^{1-1/p} \|x\|_p. \qquad \square$$

**Remark 3.4.** For $p \in (1,2)$ the $n^{1/p-1/2}$ factor in the distortion in Lemma 3.3 can be improved by using constructions of $\ell_p$ *volumetric spanners*, which give an analogue of well-conditioned spanning sets that bound the $\ell_p$ norm of the coefficients rather than the $\ell_2$ norm (Bhaskara et al., 2023). However, this requires a larger $O(n^{q/2})$ subset size $S$ and we do not have good streaming algorithms for constructing these objects. Thus, we do not incorporate this trade-off for the sake of simplicity.

### 3.2 SPACE-DISTORTION TRADE-OFFS VIA BLOCKING

By combining the results of Lemmas 3.2 and 3.3, we immediately obtain an $O(n^2 \log d)$ space algorithm with a $\text{poly}(n,d)$ distortion. To prove Theorem 7, we will now show how to apply this in a blockwise fashion to obtain a smooth trade-off between the space complexity and the distortion.

*Proof of Theorem 7.* For a block size parameter $B = \min\{\beta^q, d\} \geq n$, suppose that we run the well-conditioned spanning subset algorithm of Lemma 3.2 in each of the $O(d/B)$ blocks of $B$ columns. This algorithm stores $O(sd/B)$ columns, as we store at most $s$ columns for each of the $d/B$ blocks. Let $S$ denote the set of stored columns, and let $x^* := \arg\min_{Ax=b} \|C^{-1}x\|_p$. For the $b$-th block, let $(x^*)^b$ denote the restriction of $x^*$ to the $b$-th block of columns. Then by Lemma 3.3, $A(x^*)^b = A(\hat{x})^b$ for some $(\hat{x})^b$ that is supported on the columns stored by the algorithm and satisfies

$$\|C^{-1}(\hat{x})^b\|_p \leq \begin{cases} B^{1-1/p} \|C^{-1}(x^*)^b\|_p, & \text{when } p \geq 2, \\ n^{1/p-1/2} B^{1-1/p} \|C^{-1}(x^*)^b\|_p, & \text{when } p < 2. \end{cases}$$

By concatenating these solutions across the $O(d/B)$ blocks, we obtain a solution $\hat{x}$ that is supported on the columns stored by the algorithm, satisfies $A\hat{x} = b$, and satisfies equation 3. $\square$

## 4 LOWER BOUNDS

Due to space constraints, the proofs of most of our lower bound results (Theorems 2, 3, 4, and 5) are given in Section C. However, we give the proof of Theorem 6 here, as it is technically the most interesting and illustrates several important ideas. We first state several standard facts from information theory, which can be found in, e.g., Cover & Thomas (2001).

**Definition 4.1.** Let $X, Y, Z$ be random variables supported on sets $\mathscr{X}, \mathscr{Y}, \mathscr{Z}$ with probability laws $p_X, p_Y, p_Z$, respectively. Then, the entropy of $X$ is defined as $\mathbf{H}(X) := \sum_{x \in \mathscr{X}} p_X(x) \log_2 \frac{1}{p_X(x)}$ and the conditional entropy of $Y$ given $X$ is defined as $\mathbf{H}(Y \mid X) := \sum_{x \in \mathscr{X}, y \in \mathscr{Y}} p_X(x) \cdot p_{Y|X=x}(y) \log_2 \frac{1}{p_{Y|X=x}(y)}$. The mutual information between $X$ and $Y$ is defined as $\mathbf{I}(X;Y) = \mathbf{H}(Y) - \mathbf{H}(Y \mid X)$ and the conditional mutual information between $X$ and $Y$ given $Z$ is defined as $\mathbf{I}(X;Y \mid Z) = \mathbf{H}(Y \mid Z) - \mathbf{H}(Y \mid X, Z)$.

**Fact 4.2.** *For random variables $X$ and $Y$, we have $\mathbf{H}(X,Y) = \mathbf{H}(X) + \mathbf{H}(Y \mid X) = \mathbf{H}(Y) + \mathbf{H}(X \mid Y)$.*

**Fact 4.3** (Chain rule). *For random variables $X_1, \ldots, X_d$ and $Y$, we have $\mathbf{H}(X_1, \ldots, X_d \mid Y) = \sum_{j=1}^d \mathbf{H}(X_j \mid Y, X_{<j})$, where $X_{<j}$ denotes the random variables $\{X_{j'} : j' \in [d], j' < j\}$.*

**Fact 4.4.** *Let $X$ be a random variable supported on $\mathscr{X}$. Then, $\mathbf{H}(X) \leq \log_2 |\mathscr{X}|$, with equality achieved when $X$ distributed uniformly on $\mathscr{X}$.*

**Fact 4.5** (Data processing inequality). *Let $X, Y$ be random variables supported on $\mathscr{X}, \mathscr{Y}$ and let $f : \mathscr{Y} \to \mathscr{X}$ be a function. Let $Z = f(Y)$. Then, $\mathbf{I}(X;Y) \geq \mathbf{I}(X;Z)$.*

**Fact 4.6** (Fano's inequality). *Let $X, Y$ be random variables supported on $\mathscr{X}, \mathscr{Y}$ and let $f : \mathscr{Y} \to \mathscr{X}$ be a function. Let $\tilde{X} = f(\mathscr{Y})$ and let $E$ denote the event that $X \neq \tilde{X}$. Then, $\mathbf{H}(X \mid Y) \leq \mathbf{H}_b(\mathbf{Pr}[E]) + \mathbf{Pr}[E] \cdot \log(|\mathscr{X}| - 1)$ where $\mathbf{H}_b(x) := -x \log_2 x - (1-x) \log_2(1-x)$ is the entropy of a Bernoulli random variable with parameter $x \in [0,1]$*

Next we prove our lower bound for large approximation factors.

**Theorem 6.** *Let $p \in (1, \infty]$ and let $q = p/(p-1) = [1, \infty)$ be its Hölder conjugate exponent. Let $\beta$ be a distortion parameter such that $(\beta \log d)^{3q} = cd$ for a sufficiently small universal constant $c$. Then any randomized algorithm that computes a B-bit summary of $\boldsymbol{a} \in \{\pm 1\}^d$ from which $\hat{\boldsymbol{x}} \in \mathbb{R}^d$ can be produced such that, with probability at least $1 - 1/O(\beta \log d)^q$, we have $\langle \boldsymbol{a}, \hat{\boldsymbol{x}} \rangle = d$ and $\|\hat{\boldsymbol{x}}\|_p \leq \beta \cdot \min_{\langle \boldsymbol{a}, \boldsymbol{x} \rangle = d} \|\boldsymbol{x}\|_p$ requires $B = \Omega(d/(\beta \log d)^{2q})$.*

*Proof.* Let $\boldsymbol{a} \in \{\pm 1\}^d$ be a random sign vector. Suppose that Alice constructs a message $M$ as a function of $\boldsymbol{a}$, sends the message to Bob, and Bob constructs $\boldsymbol{x}$ such that $\langle \boldsymbol{a}, \hat{\boldsymbol{x}} \rangle = d$ with $\|\hat{\boldsymbol{x}}\|_p \leq \beta \cdot \min_{\langle \boldsymbol{a}, \boldsymbol{x} \rangle = d} \|\boldsymbol{x}\|_p$. Since we can take $\boldsymbol{x} = \boldsymbol{a}$, the optimal solution must have $\ell_p$ norm at most $d^{1/p}$.

We will lower bound the length of the message $M$ by obtaining a lower bound on the mutual information $\mathbf{I}(\hat{\boldsymbol{x}}; \boldsymbol{a})$ between Alice's random vector $\boldsymbol{a}$ and Bob's solution $\hat{\boldsymbol{x}}$. Indeed, the entropy of the message $\mathbf{H}(M)$ lower bounds the length of the message, and we have $\mathbf{I}(\hat{\boldsymbol{x}}; \boldsymbol{a}) \leq \mathbf{I}(M; \boldsymbol{a}) \leq \mathbf{H}(M)$ by the data processing inequality.

Let $\boldsymbol{y}$ be obtained by rounding the entries of $\hat{\boldsymbol{x}}$ to the nearest integer, so that $\langle \boldsymbol{a}, \boldsymbol{y} \rangle \geq d/2$ and $\boldsymbol{y}$ takes on at most $K := O(\beta \cdot d^{1/p})$ distinct values. Now consider partitioning $[K]$ into $\lceil \log_2 K \rceil$ groups, where the $\ell$-th group is given by $C_\ell = \{k \in [K] : 2^{\ell-1} \leq k \leq 2^\ell\}$. By averaging, there is a group $\ell^*$ such that $\sum_{k \in C_{\ell^*}} d_k \geq \frac{1}{\lceil \log_2 K \rceil} \sum_{k=1}^{K} d_k \geq \frac{d}{2\lceil \log_2 K \rceil}$. Note that there must be at least $2^{-\ell^*} d/(2\lceil \log_2 K \rceil)$ coordinates belonging to the class $C_{\ell^*}$, so we must have that $(2^{\ell^*-1})^p 2^{-\ell^*} \frac{d}{2\lceil \log_2 K \rceil} \leq \|\boldsymbol{y}\|_p^p \leq 2\beta^p d$ and thus we have that $2^{\ell^*} \leq L := O(\beta^q (\log d)^{q/p})$.

For each $k \in [K]$, let $G_k \subseteq [d]$ denote the subset of coordinates for which $y_j \in \{\pm k\}$, let $\boldsymbol{y}^k$ denote the restriction of $\boldsymbol{y}$ to $G_k$, and let $d_k = \langle \boldsymbol{a}, \boldsymbol{y}^k \rangle$. Let $D = \{\text{sign}(d_k)\}_{k=1}^L$. Then, $\mathbf{I}(\hat{\boldsymbol{x}}; \boldsymbol{a}) = \mathbf{I}(\hat{\boldsymbol{x}}; \boldsymbol{a} \mid D) + \mathbf{I}(\hat{\boldsymbol{x}}; D) - \mathbf{I}(\hat{\boldsymbol{x}}; D \mid \boldsymbol{a}) \geq \mathbf{I}(\hat{\boldsymbol{x}}; \boldsymbol{a} \mid D) - \mathbf{H}(D) \geq \mathbf{I}(\hat{\boldsymbol{x}}; \boldsymbol{a} \mid D) - L$. Furthermore, we have

$$\mathbf{I}(\hat{\boldsymbol{x}}; \boldsymbol{a} \mid D) \geq \mathbf{I}(\boldsymbol{y}; \boldsymbol{a} \mid D) = \sum_{j=1}^{d} \mathbf{I}(\boldsymbol{y}; a_j \mid \boldsymbol{a}_{<j}, D) = \sum_{j=1}^{d} \mathbf{H}(a_j \mid \boldsymbol{a}_{<j}, D) - \mathbf{H}(a_j \mid \boldsymbol{y}, \boldsymbol{a}_{<j}, D)$$

$$\geq \mathbf{H}(\boldsymbol{a} \mid D) - \sum_{j=1}^{d} \mathbf{H}(a_j \mid \boldsymbol{y}, D). \tag{6}$$

By Fact 4.2, the first term is bounded by $\mathbf{H}(\boldsymbol{a} \mid D) = \mathbf{H}(\boldsymbol{a}) - \mathbf{H}(D) + \mathbf{H}(D \mid \boldsymbol{a}) \geq d - L$ so it remains the bound the latter term. We do this by constructing an estimator $\hat{\boldsymbol{a}}$ for $\boldsymbol{a}$ and then applying Fano's inequality. Our estimator $\hat{\boldsymbol{a}}$ is given by $\hat{a}_j = \text{sign}(d_k) \text{sign}(y_j)$ if the $j$-th coordinate has value $|y_j| = k \leq L$, and a random sign otherwise. We will now bound $\mathbf{Pr}[\hat{a}_j \neq a_j]$.

For each $k$, let $J_k$ be the number of coordinates $j \in G_k$ such that $\hat{a}_j = a_j$. We then have $|d_k| = |\langle \boldsymbol{a}, \boldsymbol{y}^k \rangle| = k \cdot (J_k - (|G_k| - J_k)) = k \cdot (2J_k - |G_k|)$ so the probability that $\hat{a}_j = a_j$ for a coordinate $j \in G_k$ is $\frac{J_k}{|G_k|} = \frac{1}{2} + \frac{|d_k|}{2k|G_k|}$. Then, conditioned on the success of the algorithm and a choice of the $G_k$ and $d_k$, the probability that $\hat{a}_j = a_j$ is $\sum_{k=1}^{L} \frac{J_k}{d}$, which equals

$$\sum_{k=1}^{L} \frac{|G_k|}{d} \left( \frac{1}{2} + \frac{|d_k|}{2k|G_k|} \right) = \frac{1}{2} + \frac{1}{d} \sum_{k=1}^{L} \frac{|d_k|}{2k} \geq \frac{1}{2} + \frac{1}{dL} \sum_{k \in C_{\ell^*}} |d_k| \geq \frac{1}{2} + \frac{1}{2L\lceil \log_2 K \rceil} = \frac{1}{2} + \frac{1}{O(\beta \log d)^q}.$$

Then overall, we have that

$$\mathbf{Pr}[\hat{a}_j \neq a_j] \geq \left( 1 - \frac{1}{O(\beta \log d)^q} \right) \left( \frac{1}{2} + \frac{1}{O(\beta \log d)^q} \right) \geq \frac{1}{2} + \frac{1}{O(\beta \log d)^q}.$$

By Fano's inequality (Fact 4.6), we then have that $\mathbf{H}(a_j \mid \hat{a}_j) \leq \mathrm{H}_b(\mathbf{Pr}[\hat{a}_j \neq a_j]) \leq 1 - 1/O(\beta \log d)^{2q}$. Now plugging into eq. (6) gives that $\mathbf{I}(\boldsymbol{x}; \boldsymbol{a} \mid D) \geq d - L - (d - O(d/(\beta \log d)^{2q})) = \Omega(d/(\beta \log d)^{2q}) - L$ and thus $\mathbf{I}(\boldsymbol{x}; \boldsymbol{a}) \geq \Omega(d/(\beta \log d)^{2q}) - 2L = \Omega(d/(\beta \log d)^{2q})$ for $L$ small enough as required by the theorem hypothesis. Thus, the length of the message $M$ sent by Alice must be at least $\Omega(d/(\beta \log d)^{2q})$ bits, which concludes the proof. $\qquad \square$

ACKNOWLEDGMENTS

We thank the anonymous reviewers for useful feedback on improving the presentation of this work. Part of this work done while D. Woodruff was at the Simons Institute for the Theory of Computing. D. Woodruff also acknowledges a Simons Investigator Award, NSF CCF-2335412, and Office of Naval Research award number N000142112647.

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

## A  PRELIMINARIES ON ONLINE SAMPLING AND CORESETS

The literature on online sampling and coresets was initiated by the work of Cohen et al. (2016). In the setting of online coresets, the input is an insertion-only stream of $n$ vectors $\mathbf{a}_i \in \mathbb{R}^d$ for $i \in [n]$ which arrive one at a time. The goal is then to randomly select a subset of these vectors in an *online fashion*, that is, at each time step $i$, we must irrevocably decide whether to keep $\mathbf{a}_i$ or not. The selected subset is called an *online coreset* or simply a *coreset*. The work of Cohen et al. (2016) considered algorithms for outputting online coresets with the guarantee of an $\ell_2$ *subspace embedding*.

**Definition A.1.** Let $\mathbf{A} \in \mathbb{R}^{n \times d}$. Then, a linear map $\mathbf{S} \in \mathbb{R}^{r \times n}$ is an $\ell_2$ *subspace embedding for* $\mathbf{A}$ *with distortion* $(1 + \varepsilon)$ if

$$\text{for all } \mathbf{x} \in \mathbb{R}^d, \qquad \|\mathbf{SAx}\|_2 = (1 \pm \varepsilon)\|\mathbf{Ax}\|_2.$$

Note that a weighted subset of $r$ points, i.e. a coreset, can be represented by such a linear map $\mathbf{S}$ by taking the $j$-th row of $\mathbf{S}$ to be the weighted indicator for one of the $n$ points. In the offline setting, i.e., when all the $n$ vectors are known before hand as an $n \times d$ matrix $\mathbf{A}$, then $\mathbf{S}$ can be constructed as a coreset of size $\tilde{O}(\varepsilon^{-2}d)$ using a technique known as *leverage score sampling* (Drineas et al., 2006; Cohen et al., 2015). The work of Cohen et al. (2016) then showed that this technique can in fact be generalized to the online coreset setting, showing that $\mathbf{S}$ be constructed in the online setting with a size of $r = \tilde{O}(\varepsilon^{-2}d) \log \kappa^{\mathsf{OL}}$, where $\kappa^{\mathsf{OL}}$ is as defined in Definition 1.3.

In the present work, we require online constructions of $\ell_p$ subspace embeddings, rather than $\ell_2$ subspace embeddings.

**Definition A.2.** Let $\mathbf{A} \in \mathbb{R}^{n \times d}$ and $p \in [1, \infty)$. Then, a linear map $\mathbf{S} \in \mathbb{R}^{r \times n}$ is an $\ell_p$ *subspace embedding for* $\mathbf{A}$ *with distortion* $(1 + \varepsilon)$ if

$$\text{for all } \mathbf{x} \in \mathbb{R}^d, \qquad \|\mathbf{SAx}\|_p = (1 \pm \varepsilon)\|\mathbf{Ax}\|_p.$$

Analogous constructions for this setting were obtained by the work of Woodruff & Yasuda (2023a), which generalized a technique known as $\ell_p$ *Lewis weight sampling* Cohen & Peng (2015) to the online setting. We refer to Cohen et al. (2016) and Woodruff & Yasuda (2023a) for additional details.

## B  DUALITY

We prove Lemma 2.1. Consider the primal objective given by

$$\min_{\boldsymbol{Ax}=\boldsymbol{b}} \|\boldsymbol{C}^{-1}\boldsymbol{x}\|_p.$$

We write the constraint as a Lagrangian optimization problem as

$$\min_{\boldsymbol{x}\in\mathbb{R}^d} \max_{\boldsymbol{y}\in\mathbb{R}^n} \|\boldsymbol{C}^{-1}\boldsymbol{x}\|_p + \boldsymbol{y}^\top(\boldsymbol{b}-\boldsymbol{Ax}).$$

Now if $\boldsymbol{Ax} = \boldsymbol{b}$ is feasible, then by strong duality, this is equivalent to

$$\max_{\boldsymbol{y}\in\mathbb{R}^n} \min_{\boldsymbol{x}\in\mathbb{R}^d} \|\boldsymbol{C}^{-1}\boldsymbol{x}\|_p + \boldsymbol{y}^\top(\boldsymbol{b}-\boldsymbol{Ax}) = \max_{\boldsymbol{y}\in\mathbb{R}^n} \min_{\boldsymbol{x}\in\mathbb{R}^d} \|\boldsymbol{C}^{-1}\boldsymbol{x}\|_p + \boldsymbol{y}^\top\boldsymbol{b} - \boldsymbol{y}^\top\boldsymbol{Ax}.$$

Note that

$$\min_{\boldsymbol{x}\in\mathbb{R}^d} \|\boldsymbol{C}^{-1}\boldsymbol{x}\|_p - \boldsymbol{y}^\top\boldsymbol{Ax} = \min_{\boldsymbol{x}\in\mathbb{R}^d} \|\boldsymbol{C}^{-1}\boldsymbol{x}\|_p - \boldsymbol{y}^\top\boldsymbol{ACC}^{-1}\boldsymbol{x}$$

$$= \begin{cases} -\infty, & \text{when } \|\boldsymbol{CA}^\top\boldsymbol{y}\|_q > 1, \\ 0, & \text{when } \|\boldsymbol{CA}^\top\boldsymbol{y}\|_q \leq 1. \end{cases}$$

Thus, the dual problem is given by

$$\max_{\|\boldsymbol{CA}^\top\boldsymbol{y}\|_q \leq 1} \boldsymbol{y}^\top\boldsymbol{b}.$$

## C    LOWER BOUND PROOFS

### C.1    PRELIMINARIES

We will use the one-way communication lower bound for the INDEX problem. It is known that if Alice has a uniformly random binary string with $d$ bits and Bob has a uniformly random index $j \in [d]$, then in order for Bob to correctly output the $j$-th bit of Alice's string, Alice must send at least $\Omega(d)$ bits (Rao & Yehudayoff, 2020).

### C.2    LOWER BOUNDS FOR OUTPUTTING A GOOD SOLUTION

We first prove our lower bound for outputting a $(1+\varepsilon)$-factor approximate solution.

**Theorem 5.** *Let $p \in (1, \infty]$ and let $q = p/(p-1) \in [1, \infty)$ be its Hölder conjugate exponent. Let $\varepsilon \in (0, 1/(8q))$ and $d \in \mathbb{N}$. Any randomized algorithm that computes a B-bit summary of $\boldsymbol{a} \in \{\pm 1\}^d$ from which $\hat{\boldsymbol{x}} \in \mathbb{R}^d$ can be produced such that, with probability at least $2/3$, we have $\langle \boldsymbol{a}, \hat{\boldsymbol{x}} \rangle = d$ and $\|\hat{\boldsymbol{x}}\|_p \leq (1+\varepsilon) \min_{\langle \boldsymbol{a}, \boldsymbol{x} \rangle = d} \|\boldsymbol{x}\|_p$ requires $B = \Omega(d)$.*

*Proof.* By standard error reduction techniques, we may assume that the norm minimization algorithm succeeds with probability at least $9/10$. Now let $\boldsymbol{a} \in \{\pm 1\}^d$ be a random sign vector. Since we can take $\boldsymbol{x} = \boldsymbol{a}$, we obtain that

$$\min_{\langle \boldsymbol{a}, \boldsymbol{x} \rangle = d} \|\boldsymbol{x}\|_p \leq d^{1/p}.$$

Therefore, a correct $(1+\varepsilon)$-approximately optimal solution $\hat{\boldsymbol{x}}$ must have $\|\hat{\boldsymbol{x}}\|_p \leq (1+\varepsilon)d^{1/p}$. Furthermore, the number $J$ of coordinates $j \in [d]$ such that $\text{sign}(\hat{x}_j) = a_j$ satisfies

$$d = \langle \boldsymbol{a}, \hat{\boldsymbol{x}} \rangle \leq J^{1/q} \|\hat{\boldsymbol{x}}\|_p \leq (1+\varepsilon) J^{1/q} d^{1/p},$$

where the second step above follows from Hölder's inequality. Thus, $J \geq d/(1+\varepsilon)^q \geq d/(1+2q\varepsilon)$.

It follows that, for a random index $j \in [d]$, $\mathbf{Pr}[\text{sign}(\hat{x}_j) = a_j] \geq 1/(1+2q\varepsilon)$. By a union bound, it follows that if Alice, given input $\boldsymbol{a}$, were to send the algorithm's $B$-bit summary as a message to Bob, it would enable Bob to correctly solve the INDEX problem on this instance with probability at least $1 - (1 - 1/(1+2q\varepsilon)) - 1/10 \geq 2/3$. Therefore, $B = \Omega(d)$.    $\square$

**Remark C.1.** The above lower bound instance fails for $p = 1$ since for $n = 1$, we can choose an optimal 1-sparse solution by maintaining the largest element in the vector $\boldsymbol{a}$.

### C.3    LOWER BOUNDS FOR ESTIMATING THE COST

We now turn to the problem of estimating the minimum cost of the basic $\ell_p$ regression problem, i.e., problem (1). As promised, we obtain lower bounds handling the cases $p \in \{0, 1, 2\}$.

When $p = 0$, we have the following.

**Theorem 4.** *Fix $p = 0$ and take $n = 2$. Any column-arrival streaming algorithm that, with probability at least $2/3$, outputs a 2-approximation to the cost minimum cost in problem (1) requires $\Omega(d)$ bits of space.*

*Proof sketch.* It is not hard to show that for this version of the problem, there is no sublinear-space algorithm for approximating the optimal objective value to a small constant factor. Indeed, even with $n = 2$, obtaining a better than 2-approximation requires $\Omega(d)$ space, because the problem requires us to determine whether or not a vector parallel to $\boldsymbol{b}$ appears among the columns of $\boldsymbol{A}$. If the stream presents the columns of $\boldsymbol{A}$ prior to $\boldsymbol{b}$, one can then give a reduction from the standard INDEX communication problem to the (approximate) $\ell_0$-norm minimization problem.    $\square$

When $p = 1$, we show that estimating the cost of the regression problem even to a factor of $o(\sqrt{n})$ is not possible using $\text{poly}(n)$ space. This fact is intimately related to the fact that the dual of the $\ell_1$ regression problem involves an $\ell_\infty$ constraint set, and resembles a lower bound argument of Woodruff & Yasuda (2022).

We use the following result from coding theory.

**Theorem 8** (Parampalli et al. (2013)). *For any $D \geq 1$ and $n = 2^k - 1$ for some integer $k$, there exists a set $S \subseteq \{\pm 1\}^n$ and a constant $C_D$ dependent only on $D$ which satisfy (1) $|S| = n^D$, and (2) for any $s, t \in S$ such that $s \neq t$, $|\langle s, t \rangle| \leq C_D \sqrt{n}$*

Using the above result, we show the following lower bound for estimating the cost of $\ell_1$ regression.

**Theorem 3.** *Fix $p = 1$ and let $D \geq 1$ be arbitrary. There is a constant $C_D > 0$ such that any column-arrival streaming algorithm that, with probability at least $2/3$, computes an estimate $c$ with $c \leq \min_{Ax=b} \|x\|_1 < (\sqrt{n}/C_D)c$ requires $\Omega(n^D)$ bits of space. This result applies even when all entries of the input matrix $A$ lie in $\{\pm 1\}$.*

*Proof.* Let $S$ be the set constructed in Theorem 8. We will show that estimating the cost to a sufficiently small distortion solves an INDEX instance on $|S| = n^D$ items.

We associate bit strings of length $n^D$ with subsets of $S$. To solve the INDEX problem, consider the following protocol. First, Alice constructs a matrix $A$ by taking the columns to be the vectors of the subset $A \subseteq S$ associated with the input bit string. Alice then sends the message $M$ constructed from the matrix $A$ to Bob. Finally, Bob takes $b$ to be the vector of $S$ associated with the input index $j$ and uses $b$ and the message $M$ to output an estimate $c$ to the regression cost $\min_{Ax=b} \|x\|_1$.

Note that the dual problem is $\max_{\|A^\top y\|_\infty \leq 1} y^\top b$. If $b$ is an element of Alice's subset $A \subseteq S$, then for any dual feasible $y$, we have $y^\top b \leq \|A^\top y\|_\infty \leq 1$. By strong duality, there is a primal optimal solution $x$ such that $\|x\|_1 = y^\top b \leq 1$, so the optimal solution is at most 1. On the other hand, if $b$ is not an element of Alice's subset $A \subseteq S$, then we have that $\|A^\top b\|_\infty \leq C_D \sqrt{n}$ so $y = b/C_D\sqrt{n}$ witnesses a primal value of at least $\sqrt{n}/C_D$. Thus, an approximation to the cost within a factor of $\sqrt{n}/C_D$ will allow Bob to solve the INDEX problem. $\square$

When $p = 2$, which is the "easiest" case of the regression problem, we show that estimating the optimal objective value of the problem given by eq. (1) to a small constant factor requires $\Omega(n^2)$ space. Recall that $O(n^2)$ is the space required for storing and maintaining $AA^\top$: doing so enables us to compute the optimum exactly.

**Theorem 2.** *Fix $p = 2$. There is an absolute constant $\alpha > 0$ such that any column-arrival streaming algorithm that, with probability at least $2/3$, computes a $(1 + \alpha)$-approximation to the cost of problem (1) requires $\Omega(n^2)$ bits of space.*

*Proof.* We will use the fact that there exists a collection $\{P_1, \ldots, P_m\}$ of size $m \geq \exp(\Omega(n^2))$ such that (1) each $P_i$ is an $n \times n$ orthogonal projection matrix onto an $n/2$-dimensional subspace, and (2) for all $i \neq j$, we have $\|P_i - P_j\|_2 \geq \frac{1}{4}$, which is proven in Kapralov & Talwar (2013, Section 5.2) using a result of Absil et al. (2006) (see also Ghashami et al. (2016, Theorem 4.1)).

For each $i$, let $Q_i = P_i + I$. We claim that if $A$ is taken to be one of these matrices $Q_i$, then a cost-approximating algorithm must be able to tell apart the $m$ distinct choices $Q_i$ by solving the regression problem on various choices of $b$; by standard information-theoretic arguments it must therefore use $\Omega(\log m) = \Omega(n^2)$ bits of space, if it succeeds with probability at least $2/3$.

To prove this claim, we show that for all $i \neq j$, there exists a suitable vector $b$ such that the costs corresponding to $A = Q_i$ and $A = Q_j$ differ by at least some absolute constant, whereas each of these costs is at most $O(1)$. Since the optimum cost is given by $\min_{Ax=b} \|x\|_2^2 = b^\top (AA^\top)^{-1} b$, this is equivalent to showing that

$$\|(Q_i Q_i^\top)^{-1} - (Q_j Q_j^\top)^{-1}\|_2 \geq \Omega(1). \tag{7}$$

Since $P_i$ is an orthogonal projection, we can show that $(Q_i Q_i^\top)^{-1} = I - \frac{3}{4} P_i$. Indeed, let $P_i = U \Lambda U^\top$ be the eigendecomposition of $P_i$, where $\Lambda$ is a diagonal matrix with $n/2$ ones. Then $Q_i Q_i^\top = (P_i + I)(P_i + I)^\top = P_i^2 + 2P_i + I = 3P_i + I = U(3\Lambda + I)U^T$ and $(3\Lambda + I)(4I - 3\Lambda) = 12\Lambda + 4I - 9\Lambda^2 - 3\Lambda = 4I$ so by rotating back to the $U$ basis and dividing by 4, $(Q_i Q_i^\top)^{-1} = U(I - \frac{3}{4}\Lambda)U^\top = I - \frac{3}{4} P_i$. This then gives $\|(Q_i Q_i^\top)^{-1} - (Q_j Q_j^\top)^{-1}\|_2 = \|(-\frac{3}{4})(P_i - P_j)\|_2 \geq \frac{3}{4} \cdot \frac{1}{4} = \frac{3}{16}$, which establishes eq. (7) and completes the proof. $\square$