a, \hat{x} \rangle = d$ and $\|\hat{x}\|_p \le (1+\varepsilon) \min_{\langle a, x \rangle = d} \|x\|_p$ requires $B = \Omega(d)$.*

In other words, for $p > 1$, there is no algorithm using less than $d$ space that can output a $(1+\varepsilon)$-approximate solution for $\varepsilon \le 1/(8q)$.

We also obtain a lower bound in the setting of large distortions, showing that for a $\beta$-factor approximation, the streaming algorithm must use at least $\tilde{\Omega}(d/\beta^{2q})$ bits of space, provided that $\beta^{3q} \ll d$. In particular, for $p > 1$, it is in fact not possible to output a solution $x$ in poly$(n, \log d)$ space unless the approximation factor is at least poly$(d)$.

**Theorem 6.** *Let $p \in (1, \infty]$ and let $q = p/(p-1) = [1, \infty)$ be its Hölder conjugate exponent. Let $\beta$ be a distortion parameter such that $(\beta \log d)^{3q} = cd$ for a sufficiently small universal constant $c$. Then any randomized algorithm that computes a $B$-bit summary of $a \in \{\pm 1\}^d$ from which $\hat{x} \in \mathbb{R}^d$ can be produced such that, with probability at least $1 - 1/O(\beta \log d)^q$, we have $\langle a, \hat{x} \rangle = d$ and $\|\hat{x}\|_p \le \beta \cdot \min_{\langle a, x \rangle = d} \|x\|_p$ requires $B = \Omega(d/(\beta \log d)^{2q})$.*

We prove both of these lower bounds in Section 4. The high-accuracy lower bound of Theorem 5 follows from a relatively simple reduction to one-pass streaming lower bounds for the INDEX communication problem (Kremer et al., 1995). On the other hand, Theorem 6 is our most technically advanced lower bound result, requiring additional techniques in order to extract information about $a$ from a $\beta$-approximate solution for large $\beta$. In particular, our lower bound argument involves classifying the entries of a $\beta$-approximate solution $x$ according to their contribution towards partitioning

the coordinates into comparable classes. We then apply conditioning on an additional short string of advice to construct an estimator that extracts many bits of information about the input $a$ from the solution $x$.

Turning to upper bounds, we give a new algorithm for the general $\ell_p$ regression problem (2) that runs in a strongly sublinear $d^{1-\Omega(1)}$ amount of space and outputs a solution vector achieving $d^{1-\Omega(1)}$ distortion. Furthermore, in the case of $p = 1$, our algorithm outputs a solution with distortion at most $\sqrt{n}$ with space complexity $\mathrm{poly}(n, \log d)$, which explains why the lower bounds of Theorems 5 and 6 do not apply when $p = 1$.

Below, we write $\boldsymbol{A}|^S$ to denote the $n \times |S|$ submatrix of $\boldsymbol{A}$ with columns indexed by $S \subseteq [d]$.

**Theorem 7.** *Let $p \in [1, \infty]$ and let $q = p/(p-1) \in [1, \infty]$ be its Hölder conjugate exponent. There is an algorithm that reads $\boldsymbol{A} \in \mathbb{R}^{n \times d}$ and the diagonal matrix $\boldsymbol{C} \in \mathbb{R}^{d \times d}$ in a column-arrival stream and stores a subset $S$ of at most $O(sd/\beta^q)$ columns of $\boldsymbol{A}$ and entries of $\boldsymbol{C}$ for $p > 1$, and $O(s)$ columns of $\boldsymbol{A}$ and entries of $\boldsymbol{C}$ for $p = 1$, such that*

$$\min_{\boldsymbol{A}|^S \boldsymbol{x} = \boldsymbol{b}} \|\boldsymbol{C}^{-1}\boldsymbol{

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

 $\mathrm{supp}(\boldsymbol{z}) \subseteq S$, $\boldsymbol{A}\boldsymbol{x} = \boldsymbol{A}\boldsymbol{z}$, and*

$$\|\boldsymbol{z}\|_p \le \begin{cases} d^{1-1/p}\|\boldsymbol{x}\|_p, & \text{when } p > 2, \\ n^{1/p-1/2} d^{1-1/p}\|\boldsymbol{x}\|_p, & \text{when } p \le 2. \end{cases}$$

*Proof.* We may assume that $d \ge n$, since otherwise we can take $\boldsymbol{z} = \boldsymbol{x}$. By the definition of well-conditioned spanning subsets, we can write $\boldsymbol{A} = (\boldsymbol{A}\boldsymbol{C})|^S (\boldsymbol{C}|^S)^{-1} \boldsymbol{Y} = \boldsymbol{A}|^S \boldsymbol{Y}$ where $\boldsymbol{A}|^S$ denotes the

columns of $\boldsymbol{A}$ indexed by $S$ and $\boldsymbol{Y}$ is an $|S| \times d$ matrix of coefficients with $\|(\boldsymbol{C}|^S)^{-1}\boldsymbol{Y}\boldsymbol{e}^{(j)}\|_2^2 \leq 1$ for every column $j \in [d]$. Then, $\boldsymbol{A}\boldsymbol{x} = \boldsymbol{A}|^S\boldsymbol{Y}\boldsymbol{x}$ so padding the vector $\boldsymbol{Y}\boldsymbol{x}$ with zeros gives our vector $\boldsymbol{z}$ that is supported on $S$. Furthermore,

$$\|\boldsymbol{C}^{-1}\boldsymbol{z}\|_p = \|(\boldsymbol{C}|^S)^{-1}\boldsymbol{Y}\boldsymbol{x}\|_p \leq \sum_{j=1}^d \|(\boldsymbol{C}|^S)^{-1}\boldsymbol{Y}\boldsymbol{

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

_{\mathbf{x} \in \mathbb{R}^d} \max_{\mathbf{y} \in \mathbb{R}^n} \|\mathbf{C}^{-1}\mathbf{x}\|_p + \mathbf{y}^\top (\mathbf{b} - \mathbf{Ax}).$$

Now if $\mathbf{Ax} = \mathbf{b}$ is feasible, then by strong duality, this is equivalent to

$$\max_{\mathbf{y} \in \mathbb{R}^n} \min_{\mathbf{x} \in \mathbb{R}^d} \|\mathbf{C}^{-1}\mathbf{x}\|_p + \mathbf{y}^\top (\mathbf{b} - \mathbf{Ax}) = \max_{\mathbf{y} \in \mathbb{R}^n} \min_{\mathbf{x} \in \mathbb{R}^d} \|\mathbf{C}^{-1}\mathbf{x}\|_p + \mathbf{y}^\top \mathbf{b} - \mathbf{y}^\top \mathbf{Ax}.$$

Note that

$$\min_{\mathbf{x} \in \mathbb{R}^d} \|\mathbf{C}^{-1}\mathbf{x}\|_p - \mathbf{y}^\top \mathbf{Ax} = \min_{\mathbf{x} \in \mathbb{R}^d} \|\mathbf{C}^{-1}\mathbf{x}\|_p - \mathbf{y}^\top \mathbf{A}\mathbf{C}\mathbf{C}^{-1}\mathbf{x}$$
$$= \begin{cases} -\infty, & \text{when } \|\mathbf{C}\mathbf{A}^\top \mathbf{y}\|_q > 1, \\ 0, & \text{when } \|\mathbf{C}\mathbf{A}^\top \mathbf{y}\|_q \le 1. \

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

 $\boldsymbol{b}$ such that the costs corresponding to $\boldsymbol{A} = \boldsymbol{Q}_i$ and $\boldsymbol{A} = \boldsymbol{Q}_j$ differ by at least some absolute constant, whereas each of these costs is at most $O(1)$. Since the optimum cost is given by $\min_{\boldsymbol{Ax}=\boldsymbol{b}} \|\boldsymbol{x}\|_2^2 = \boldsymbol{b}^\top (\boldsymbol{A}\boldsymbol{A}^\top)^{-1}\boldsymbol{b}$, this is equivalent to showing that

$$\|(\boldsymbol{Q}_i\boldsymbol{Q}_i^\top)^{-1} - (\boldsymbol{Q}_j\boldsymbol{Q}_j^\top)^{-1}\|_2 \geq \Omega(1). \tag{7}$$

Since $\boldsymbol{P}_i$ is an orthogonal projection, we can show that $(\boldsymbol{Q}_i\boldsymbol{Q}_i^\top)^{-1} = \boldsymbol{I} - \frac{3}{4}\boldsymbol{P}_i$. Indeed, let $\boldsymbol{P}_i = \boldsymbol{U}\boldsymbol{\Lambda}\boldsymbol{U}^\top$ be the eigendecomposition of $\boldsymbol{P}_i$, where $\boldsymbol{\Lambda}$ is a diagonal matrix with $n/2$ ones. Then $\boldsymbol{Q}_i\boldsymbol{Q}_i^\top = (\boldsymbol{P}_i + \boldsymbol{I})(\boldsymbol{P}_i + \boldsymbol{I})^\top = \boldsymbol{P}_i^2 + 2\boldsymbol{P}_i + \boldsymbol{I} = 3\boldsymbol{P}_i + \boldsymbol{I} = \boldsymbol{U}(3\boldsymbol{\Lambda} + \boldsymbol{I})\boldsymbol{U}^T$ and $(3\boldsymbol{\Lambda} + \boldsymbol{I})(4\boldsymbol{I} - 3\boldsymbol{\Lambda}) = 12\boldsymbol{\Lambda} + 4\boldsymbol{I} - 9\boldsymbol{\Lambda}^2 - 3\boldsymbol{\Lambda} = 4\boldsymbol{I}$ so by rotating back to the $\boldsymbol{U}$ basis and dividing by 4, $(\boldsymbol{Q}_i\boldsymbol{Q}_i^\top)^{-1} = \boldsymbol{U}(\boldsymbol{I} - \frac{3}{4}\boldsymbol{\Lambda})\boldsymbol{U}^\top = \boldsymbol{I} - \frac{3}{4}\boldsymbol{P}_i$. This then gives $\|(\boldsymbol{Q}_i\boldsymbol{Q}_i^\top)^{-1} - (\boldsymbol{Q}_j\boldsymbol{Q}_j^\top)^{-1}\|_2 = \|(-\frac{3}{4})(\boldsymbol{P}_i - \boldsymbol{P}_j)\|_2 \geq \frac{3}{4} \cdot \frac{1}{4} = \frac{3}{16}$, which establishes eq. (7) and completes the proof. $\square$