# OpenReview forum: "Streaming Algorithms For $\ell_p$ Flows and $\ell_p$ Regression"
_ICLR.cc/2025/Conference — ICLR 2025 Spotlight_

### Official Review · Reviewer_fogf · 2024-10-20

**Soundness:** 3
**Presentation:** 3
**Contribution:** 3
**Rating:** 8
**Confidence:** 4

**Summary:**

The $\ell_p$ regression $\min_{\mathbf{Ax} = \mathbf{b}} ||\mathbf{x}||_p$ is a fundamental problem in machine learning, data science, and numerical linear algebra. When $p = 2$, it is the classical linear regression problem and has a closed-form solution $\mathbf{x}^* = \mathbf{A}^\top (\mathbf{A}\mathbf{A}^\top)^{-1} \mathbf{b}$. The general $\ell_p$ regression has been well-studied, in particular, when $\mathbf{A}$ is the vertex-edge incidence matrix, it is the $\ell_p$-norm flow problem, which corresponds to transshipment, Laplacian solver, and maximum flow if $p = 1, 2$, and $\infty$ respectively. This paper considers the scenario that the matrix $\mathbf{A} \in \mathbb{R}^{n \times d}$ has $n \ll d$, which leads to the linear system $\mathbf{Ax} = \mathbf{b}$ being underdetermined. In this case, it is impractical to store all the $d$ columns of $\mathbf{A}$, which motivates the usage of the column-arrival streaming model. In addition, there are two kinds of solutions to $\ell_p$ regression: (1) $\textit{cost estimation}$ that approximates the optimal $\ell_p$ norm; (2) $\textit{vector-valued}$ that approximates the optimal solution $\mathbf{x}^*$. For the first kind, this paper achieves space complexity $\widetilde{O}(\epsilon^{-2} n^{\ 1+max\\{1, p/2(p-1)\\}})$, which is $\widetilde{O}(\epsilon^{-2} n^2)$ when $p > 2$, and also gives lower bounds for $p = 0, 1, 2$. For the second kind, this paper also provides lower bounds and upper bounds for different values of $p$.

**Strengths:**

(1) This paper researches the general $\ell_p$ regression problem under the column-arrival streaming model, while the existing work only considers the special case, like $p = 2$ or $\mathbf{A}$ is the vertex-edge incidence matrix.
(2) For general $p$, this paper establishes both lower and upper bounds on space complexity for the two kinds of solutions: cost estimation and vector-valued problems. These new results strengthen the paper's contributions.
(3) This paper has an extensive set of references and a comprehensive summary of the related work.

**Weaknesses:**

The $p$-norm flow problem defined in (1) (Line 61) is not exact. The correct one should be $\min_{\mathbf{Ax} = \mathbf{b}} ||\text{diag}(\mathbf{w}) \cdot \mathbf{x}||_p$. To be specific, when $p = 1$ and $\mathbf{w}$ is the cost vector, then it is a transshipment problem; when $p = 2$ and $\mathbf{w} = \mathbf{r}^{1/2}$, where $\mathbf{r}$ is the resistance vector, then it is reduced to a Laplacian solver; when $p = \infty$ and $\mathbf{w} = \mathbf{c}^{-1}$, where $\mathbf{c}$ is the capacity vector, then it is the maximum flow problem.

By this formulation, it is the same as formulation (2). As a special case, when $\mathbf{A}$ is a vertex-edge incidence matrix in the graph streaming problem, there is some existing work (Line 91-97) on transshipment, electrical flow, maximum flow, etc. Could you compare this work with their results?

**Questions:**

(1) In Theorem 1, Line 153, should "$s$ rows" be "$s$ columns"? At Line 164, should the second $\widetilde{O}(\epsilon^{-2} n)$ be $\widetilde{O}(\epsilon^{-2} n^2)$? Please verify that.
(2) In Theorem 1, when $p \to 1$, like $p = 1.1$, $q = p/(p-1)$ would be a large constant. In this case, could you discuss the behavior of the proposed algorithm? If it is indeed a non-trivial issue, could you give a brief analysis or comment in this paper on $p \to 1$?

---

> ### Author Response · Authors · 2024-11-27
>
> Weaknesses: Agreed, equation (2) is meant to capture the general setting when the Lp norms are weighted and capture the cases which you mention. Surprisingly, there are in fact not many algorithmic results known for streaming flow problems. The works mentioned in 91-97 are lower bounds against the directed version of the problem, which are hard for streaming, whereas we present new algorithms for the undirected version. Known results are presented in Table 1.
>
> (1) Thank you for catching the typo for "rows" vs "columns", this has been fixed. In the graph setting, the columns are sparse so the number of bits is asymptotically the same as the number of columns.
> (2) Indeed, if p goes to 1, there are inherent difficulties, as we discuss in 187 and complement with a lower bound in Theorem 3.

---

> > ### Comment · Reviewer_fogf · 2024-12-02
> >
> > Thank the authors for your feedback! I will retain my score.

---

### Official Review · Reviewer_SDMB · 2024-10-29

**Soundness:** 4
**Presentation:** 4
**Contribution:** 4
**Rating:** 8
**Confidence:** 5

**Summary:**

The paper initiates the study of $p$-norm regression in the one-pass streaming setting. They consider the underdetermined setting $\min_{Ax=b} \|x\|_p$ where $A$ has size $n\times d$, $n<<d$, receives $d$ column updates. They consider $p\in [1,\infty)$. When we consider instances graphs, these updates correspond to edge insertions.

The first result is an algorithm for “p-norm flow sparsifier” in $O(n^2)$ space which is a general version of graph sparsifiers which minimize the $\ell_p$-norm. This algorithm follows from using the online lewis weight sampling algorithm of Woodruff and Yasuda’22 on the dual problem. The other algorithmic result deals with giving a tradeoff between the size of the sparsifier and objective error when the goal is to maintain the entire solution vector x. The first algorithm only maintains the objective value.

The paper also presents several information theoretic lower bounds which match these upper bounds. They also give some extra lower bounds for the case of $p=2$.

**Strengths:**

The paper studies an important problem and makes a significant amount of progress in the new direction of streaming algorithms for regression. They also leave some good open questions in the paper. The techniques are also quite simple and use previous works quite well. The main technical core is the lower bound part, which is of independent interest.

**Weaknesses:**

It seems like there are too many ideas and previous results used in the paper. Would be useful to have some more preliminaries in the appendix or somewhere.

**Questions:**

1. Can the authors add a comparison with what is known in online algorithms in the corresponding settings?
2. The tables in the paper, that give the lower and upper bounds for different settings, can these be split in a different way so that its easier to contrast the lower and upper bounds for all settings? This would also make clear what settings are still unknown.

---

> ### Author Response · Authors · 2024-11-27
>
> > It seems like there are too many ideas and previous results used in the paper. Would be useful to have some more preliminaries in the appendix or somewhere.
>
> We have added a preliminaries section in the appendix with the most important definitions and if the reviewer has any further suggestions as to what would be helpful, we would be happy to add them.
>
> 1. We have added a discussion on the work of https://arxiv.org/abs/1604.05448, which is the only other work we are aware of on online graph sparsification.
> 2. The tables in the current draft are organized as Table 1 for algorithms and lower bounds for estimating the cost, and Table 2 for algorithms and lower bounds for outputting the solution. We are happy to take suggestions on what other organization would be more clear to readers.

---

### Official Review · Reviewer_Rniw · 2024-11-05

**Soundness:** 4
**Presentation:** 4
**Contribution:** 3
**Rating:** 8
**Confidence:** 2

**Summary:**

The paper gives upper bounds and lower bounds for streaming underdetermined $\ell_p$ linear regression in the column arrival model. The results apply both to the value of the best fit, and to the actual solution. There are various results, depending on the value of $p$ and the approximation guarantees. The arguments use duality and sparsification.

**Strengths:**

The paper explores the full range of values of $p$.

**Weaknesses:**

The methods rely substantially on past work.

**Questions:**

None

---

### Meta-Review · Area_Chair_YUaP · 2024-12-19

**Metareview:**

This paper provides streaming algorithms for solving linear regression problems with an $\ell_p$ constraint. In the streaming model data arrives one coordinate at a time. The reviewers deem the results to be good, and this is definitely publishable.

In my own reading, this seems to be an algorithms focused paper, and may not have the proper/big audience in ICLR. However, this is no fault of the authors. The reviews have also been short, but positive.

Overall I recommend acceptance.

**Additional Comments On Reviewer Discussion:**

The reviews have been positive and uniform; little issues raised, and the authors provided precise response.

---

### Decision · Program_Chairs · 2025-01-22

Accept (Spotlight)